# Molecular Dynamics Insight into the CO_2_ Flooding Mechanism in Wedge-Shaped Pores

**DOI:** 10.3390/molecules28010188

**Published:** 2022-12-26

**Authors:** Lu Wang, Weifeng Lyu, Zemin Ji, Lu Wang, Sen Liu, Hongxu Fang, Xiaokun Yue, Shuxian Wei, Siyuan Liu, Zhaojie Wang, Xiaoqing Lu

**Affiliations:** 1College of Science, China University of Petroleum, Qingdao 266580, China; 2State Key Laboratory of Enhanced Oil Recovery, Research Institute of Petroleum Exploration & Development, CNPC, Beijing 100083, China; 3School of Materials Science and Engineering, China University of Petroleum, Qingdao 266580, China

**Keywords:** CO_2_ flooding, wedge-shaped pores, transport mechanism, molecular dynamics simulation

## Abstract

Because of the growing demand for energy, oil extraction under complicated geological conditions is increasing. Herein, oil displacement by CO_2_ in wedge-shaped pores was investigated by molecular dynamics simulation. The results showed that, for both single and double wedge-shaped models, pore Ⅱ (pore size from 3 to 8 nm) exhibited a better CO_2_ flooding ability than pore Ⅰ (pore size from 8 to 3 nm). Compared with slit-shaped pores (3 and 8 nm), the overall oil displacement efficiency followed the sequence of 8 nm > double pore Ⅱ > single pore Ⅱ > 3 nm > double pore Ⅰ > single pore Ⅰ, which confirmed that the exits of the wedge-shaped pores had determinant effects on CO_2_ enhanced oil recovery over their entrances. “Oil/CO_2_ inter-pore migration” and “siphoning” phenomena occurred in wedge-shaped double pores by comparing the volumes of oil/CO_2_ and the center of mass. The results of the interaction and radial distribution function analyses indicate that the wide inlet and outlet had a larger CO_2_–oil contact surface, better phase miscibility, higher interaction, and faster displacement. These findings clarify the CO_2_ flooding mechanisms in wedge-shaped pores and provide a scientific basis for the practical applications of CO_2_ flooding.

## 1. Introduction

The excessive consumption of oil resources has been increasing with the rapid growth of the world economy and society [1,2]. To meet this growing demand [3], the development of unconventional resources such as micro- and nano-scale reservoirs has gradually attracted people’s attention [4,5,6]. At present, the most frequently utilized methods for oil displacement are gas flooding, hot water flooding, and surfactant flooding. However, compared with gas flooding, hot water flooding is accompanied by the enormous investment of ground facilities, high heat injection costs, and environmental pollution [7]. Similarly, surfactant flooding uses a lot of chemical solvents, which raises the cost of manufacturing and causes serious damage to reservoirs [8]. Gas flooding technologies using N_2_ [9], CH_4_ [10], CO_2_ [11], and other gaseous oil [12] are commonly used. Among these, supercritical CO_2_ can be easily dissolved in alkanes [13], which expands the volume of oil [14], decreases the viscosity [15], and declines the gas–liquid surface tension [16]. CO_2_ flooding is more versatile and has a higher recovery rate than other techniques. It is not only applicable to traditional oil fields, but also clearly improves oil recovery in reservoirs with extremely low permeability. Additionally, it can function as an efficient geological CO_2_ sequestration to solve issues caused by the greenhouse effect [17]. Current investigations of the mechanism for CO_2_ flooding in micro- and nano-scale pores is focused mainly on single slit-shaped pores. However, the different environments of oil reservoirs may lead to complexity of the pore structures [18,19,20,21,22], in which the reservoir storage capacity and fluid percolation characteristics are not the same due to the different throat distribution morphology and connectivity [4,23,24,25]. Fang et al. [26] and Lu et al. [27] constructed a rough surface on slit-shaped pores and found that the grooves or bulges on the solid surface had a significant influence on the fluid transport behavior, proving that the roughness of micro- and nano-pore structure would greatly affect the recovery of the crude oil. Mohammad Sejgi et al. [28] studied the multiphase fluid/fluid displacement in nanoscale pores with different cross-sectional shapes, including circular, square, and right-angled triangle. It was found that the angular cross section in the nanopore affects the fluid transport. Studies have indicated that a large percentage of wedge-shaped pores exist in many areas, such as shale oil reservoirs in the Bohai Bay basin [29]; the salt lake basin sedimentary strata of the Jimsar depression in Xinjiang [24]; and the East Morgan oil field in the Gulf of Suez, Egypt [30]. However, the oil displacement in wedge-shaped pores and related mechanisms, including oil/CO_2_ transport process, phase miscibility, CO_2_–oil–rock interaction, and displacement efficiency, have not been systematically elucidated.

Molecular dynamics simulation has been widely used in the exploration of CO_2_ flooding in micro- and nano-scale reservoirs [31,32]. Therefore, it was applied herein to investigate the microscopic behavior of fluid during CO_2_ flooding in wedge-shaped quartz nanopores. Firstly, we simulated the transport process in wedge-shaped pores during oil displacement. Then, we investigated the oil/CO_2_ inter-pore migration mechanism and the restriction effect from the inlet and outlet. Subsequently, we studied the microscopic CO_2_–oil–rock interactions and miscible phases during CO_2_ flooding in wedge-shaped pores. Finally, the oil displacement in wedge-shaped pores was compared with that in slit-shaped pores to elucidate the effect of wedge-shaped pores on oil recovery. 

## 2. Results and Discussion 

Here, four types of wedge-shaped pores, including single pore Ⅰ with a pore size from 8 to 3 nm, single pore Ⅱ with a pore size from 3 to 8 nm, and double pores including double pore Ⅰ and double pore Ⅱ, are explored. Two types of slit-shaped pores, 3 nm and 8 nm pores, are also considered as a comparison to evaluate the effect of wedge-shaped pores on CO_2_ flooding. 

### 2.1. CO_2_ Flooding Process

Snapshots of the oil displacement process during CO_2_ injection into wedge-shaped pores are shown in Figure 1. As seen in Figure 1a, initially, single pore Ⅰ exhibits a large contact area of CO_2_ with the oil at the inlet, and some oil molecules diffuse into the left side of the pore, and oil molecules then re-enter wedge-shaped pores together with CO_2_. Before 1.5 ns, CO_2_ diffuses along the rock surface in wedge-shaped pores. At 2.5 ns, a small number of CO_2_ molecules pass through the pore and excrete from the 3 nm outlet. At 3.5 ns, the space of the wedge-shaped pores is mostly occupied by CO_2_, and after 4.5 ns, the oil displacement is nearly complete. As seen in Figure 1b, single pore Ⅱ has a small pore size at the inlet, and no oil molecules diffuse into the left side of single pore Ⅱ over time. The contact between CO_2_ and oil molecules is comparatively small at the initial stage, but it turns larger after CO_2_ molecules enter the pore. After 1.5 ns, a small amount of CO_2_ can pass through the pore, and more CO_2_ molecules enter single pore Ⅱ compared with those in single pore Ⅰ after 2.5 ns. The oil is almost entirely displaced after 3.5 ns, and the whole CO_2_ flooding process is much faster than that in single pore Ⅰ. 

Figure 1c shows that the transport phenomena in double pores are similar to those in the two single pores. The transport speed of each phase is roughly the same at the initial state. After 1.5 ns, the transport speed of CO_2_ to the right side of double pore Ⅰ is much slower than that in single pore Ⅰ. More obviously, the transport of CO_2_ in double pore Ⅱ is much faster than that in double pore Ⅰ, and this difference is larger than that between single pore Ⅰ and single pore Ⅱ. This phenomenon may be ascribed to the fact that a small portion of oil molecules together with CO_2_ in double pore Ⅰ can enter double pore Ⅱ and eventually be displaced from double pore Ⅱ. 

### 2.2. Oil/CO_2_ Inter-Pore Migration Mechanism

To quantitatively describe the CO_2_ flooding process, the number of CO_2_ molecules and the ratio of residual oil in wedge-shaped pores during the displacement processes are provided in Figure 2. Figure 3 presents the number of oil molecules on the left side of double pore Ⅰ and single pore Ⅰ, and the number of oil molecules migrating between double pore Ⅰ and double pore Ⅱ. Accordingly, the mechanism of oil/CO_2_ inter-pore migration is proposed in Figure 4.

Figure 2a shows that the amount of CO_2_ entering double pore Ⅰ is roughly the same as that in single pore Ⅰ throughout the whole displacement. Before 1.5 ns, the number of CO_2_ entering double pore Ⅱ is very close to that of single pore Ⅱ; from 1.5 ns to 2.5 ns, the number of CO_2_ molecules entering double pore Ⅱ is slightly larger than that of single pore Ⅱ; from 2.5 ns to 4.5 ns, the number of CO_2_ molecules in double pore Ⅱ is slightly smaller than that of single pore Ⅱ. Because of the same pore shape, CO_2_ molecules will fill up the whole pore space for both single pore Ⅱ and double pore Ⅱ eventually, and the number of CO_2_ molecules will reach to the same value. The displacement rate is higher in single pore Ⅱ and double pore Ⅱ than that in single pore Ⅰ and double pore Ⅰ due to the influence of the pore structure. The number of CO_2_ molecules entering single pore Ⅱ and double pore Ⅱ at the same time is greater than that entering single pore Ⅰ and double pore Ⅰ. As seen from Figure 2b, not much difference can be seen between the ratio of the remaining oil in single pore Ⅰ and double pore Ⅰ, while the displacement rate is slightly larger in double pore Ⅰ. The ratio of residual oil in double pore Ⅱ is smaller than that in single pore Ⅱ, and the displacement rate in double pore Ⅱ is faster than that in single pore Ⅱ.

Figure 3a shows the number of oil molecules on the left side of double pore Ⅰ and single pore Ⅰ. It indicates that there were less oil molecules on the left side of double pore Ⅰthan that of single pore Ⅰ. The number of oil molecules migrating between these two double pores is compared in Figure 3b. Evidently, more oil molecules in double pore Ⅰ migrate into double pore Ⅱ, while very few oil molecules in double pore Ⅱ migrate into double pore Ⅰ.

By observing the trajectory of CO_2_, we plotted the streamlines of CO_2_ fluid near the two pores to directly reflect the molecular migration in Figure 4. When free phase CO_2_ approached the interlayer rock, some of free phase CO_2_ in double pore Ⅰnear the rock between the pores travelled towards double pore Ⅱ along the rock surface due to the obstruction of the wedge-shaped rocks, and a “CO_2_ inter-pore migration” phenomenon occurred. This explains why the number of CO_2_ molecules entering double pore Ⅱ was slightly higher than that in single pore Ⅱ at 1.5 ns to 2.5 ns. In Figure 4, the trajectory of oil molecules in double pore Ⅰ is plotted. In double pore Ⅰ, the oil molecules were mainly divided into two parts: most of the oil molecules were displaced from the right outlet due to the pressure difference, and a small part of the oil molecules diffused to the left side of the pore. This was due to the oil molecules in double pore Ⅰ being hindered by the pore structure, which prevented some oil molecules migrating into the right side. In addition, there was a pressure difference between the inlet and outlet due to the fast flow rate and low pressure at the inlet and the slow flow rate and high pressure at the outlet. However, because steady pressure was applied to the left He plate, only a small percentage of the oil molecules could migrate to the left. Combined with the results in Figure 2, most of oil molecules were displaced in the late stage of displacement together with a constant number of CO_2_ molecules in double pore Ⅱ. The ratio of residual oil in double pore Ⅱ was lower than that in single pore Ⅱ, and the number of CO_2_ molecules in double pore Ⅱ was less than that in single pore Ⅱ. This is because the oil displacement efficiency in double pore Ⅱ was larger than that in double pore Ⅰ, so the fluid density inside double pore Ⅱ decreased and the pressure weakened. As a result, the pressure difference between these two double pores causes inter-pore migration of oil from double pore Ⅰ into double pore Ⅱ together with CO_2_ injection, thus creating a phenomenon similar to "siphoning". Both the flow of oil molecules in Figure 1c and the number of oil molecules migrating from double pore Ⅰ to Ⅱ in Figure 3b clearly demonstrate the oil inter-pore migration mechanism, as exhibited in Figure 4.

### 2.3. Interaction 

The interaction energies between oil–rock and oil–CO_2_ were calculated to reflect the competitive adsorption and transport process of oil–CO_2_ molecules on the rock surface. Equation (1) is used to calculate the interaction energy between oil and rock surface [33], and Equation (2) is used to calculate the interaction energy between oil and CO_2_ [34]:(1)$$E_{oil-rock}=E_{oil+rock}-\left(E_{oil}+E_{rock}\right)$$
(2)$$E_{oil-CO_{2}}=E_{oil+CO_{2}}-\left(E_{oil}+E_{CO_{2}}\right)$$
where $E_{oil-rock}$ and $E_{oil-CO_{2}}$ are the interaction energies between oil–rock and oil–CO_2_; $E_{oil+rock}$ and $E_{oil+CO_{2}}$ are the energies of the systems including oil–rock and oil–CO_2_; and $E_{oil}$, $E_{rock}$, and $E_{CO_{2}}$ represent the energy of oil molecules, rock surface, and CO_2_ molecules, respectively. It can be seen from Figure 5, as a whole, $E_{oil-rock}$ gradually decreased with time, and $E_{oil-CO_{2}}$ gradually increased in both the slit-shaped and wedge-shaped pores. These results indicate that CO_2_ gradually stripped oil from the rock surface during the displacement process [35], the contact between CO_2_ and oil increased and the driving force from CO_2_ was strengthened.

Comparing $E_{oil-rock}\;$ in different pores, it was found that the $E_{oil-rock}$ in 3 nm slit-shaped pore decreased more slowly than that in the 8 nm slit-shaped pore. In wedge-shaped pores, $E_{oil-rock}$ decreased to 0 Kcal/mol in single pore Ⅰ and double pore Ⅰ simultaneously, indicating that CO_2_ stripped off oil along the rock surface at the same rate. Comparatively, $E_{oil-rock}$ in single pore Ⅰ and double pore Ⅰ decreased more slowly than that in single pore Ⅱ and double pore Ⅱ. This suggests that the wide pore outlet resulted in a faster oil stripping by CO_2_ than the narrow pore outlet. 

In the slit-shaped pores, $E_{oil-CO_{2}}$ gradually increased with time, and the tendency became more evident with the increase in the interfacial molecule number of CO_2_–oil, as shown in Figure 5b. The tendency of $E_{oil-CO_{2}}$ increased sharply in the early displacement stage and then slowed down. Comparatively, $E_{oil-CO_{2}}$ exhibited a linear growth in single pore Ⅱ and double pore Ⅱ throughout the oil displacement process, because the pore diameter gradually increased and the contact between CO_2_ and oil kept increasing. In addition, because single pore Ⅰ and single pore Ⅱ contained the same number of oil molecules, the $E_{oil-CO_{2}}$ for both pores eventually reached the same value, as shown in Figure 5b. $E_{oil-CO_{2}}$ in double pore Ⅰ was larger than that in double pore Ⅱ in the latter displacement stage. This originated from the migrated oil molecules from double pore Ⅰ into double pore Ⅱ, which occupied the pore space that should have belonged to CO_2_.

### 2.4. Effect of Pore Shape on CO_2_–Oil Miscibility

The CO_2_–oil miscibility had a great influence on the displacement process during the entire oil recovery. In wedge-shaped pores, the miscible phase zone formed via three processes, including the (1) migration of separated oil molecules into CO_2_, (2) the dissolution of bulk oil molecules in CO_2_, and the (3) penetration of CO_2_ into bulk oil, which is in good agreement with those in slit-shaped pores [36]. The radial distribution function (RDF) g(r) between C (CO_2_)-C (oil) atoms is presented in Figure 6 to describe the distribution of CO_2_ around oil. According to the snapshots in Figure 1 and RDF in Figure 7a–d, the peak and the area below the curve of RDF in the single pore Ⅰ and double pore Ⅰ were larger than those in the single pore Ⅱ and double pore Ⅱ in the early stage of displacement. This was due to the larger contact surface between CO_2_ and oil in single pore Ⅰ and double pore Ⅰ in the early stage of displacement, and the restriction at the outlet made the oil molecules less likely to be displaced out. These two factors led to a better mixing of CO_2_ and oil in single pore Ⅰ and double pore Ⅰ, which is in good agreement with the interaction analyses. The curves of RDF in single pore Ⅰ and double pore Ⅰ became denser, indicating that there was decrease in miscibility tendency, as well as the dissolution of CO_2_ and oil. The g(r) value of double pore Ⅰ was still larger than that of double pore Ⅱ because part of the oil diffused to the left side of the double pores with a large diffusion and more contact with CO_2_.

### 2.5. Effect of Wedge-Shaped Pore on Oil Recovery

To better understand the oil recovery in wedge-shaped pores, the center of mass (COM) of oil molecules during the transport process was calculated. According to Figure 7a, COM changed more quickly in single pore Ⅱ and double pore Ⅱ than that in single pore Ⅰ and double pore Ⅰ, which was constrained by pores shape. In addition, the transport of oil molecules was faster, and the displacement efficiency was higher in double pores than that in single pores. The transport of oil molecules was accelerated in the double pore due to the "siphoning" phenomenon.

By comparing the crude oil recovery in Figure 7b, we can see that the displacement efficiency in wedge-shaped double pore Ⅱ was the highest, followed by single pore Ⅱ > double pore Ⅰ > single pore Ⅰ. This means the oil displacement efficiency would be improved in wedge-shaped double pores with respect to their corresponding single pores. To estimate the effect of wedge-shaped pores on oil recovery, we compared the displacement in wedge-shaped single pores with that in slit-shaped pores. As a whole, the efficiency of the displacement process followed the sequence of 8 nm > double pore Ⅱ > single pore Ⅱ > 3 nm > double pore Ⅰ > single pore Ⅰ. Compared with the 3 nm slit-shaped pores, a wedge-shaped single pore with an enlarged outlet size would be facile to the enhancement of oil recovery, while a wedge-shaped single pore with a larger inlet size would be unfavorable for the enhancement of oil recovery. Compared with the 8 nm slit-shaped pores, both the oil recovery and displacement rate would sharply decrease whether the inlet size decreased or outlet size decreased. In addition, the mean square displacement (MSD) in Appendix A shows that the fluidity in single pore Ⅱ and double pore Ⅱ was better than that in single pore Ⅰ and double pore Ⅰ, giving good support regarding the effect of wedge-shaped pores for oil recovery.

## 3. Models and Methodology

### 3.1. Modeling

The silica surface was chosen as the typical mineral composition and the α-square quartz crystalline surface along the (010) crystallographic direction was cut. One H atom was attached to each O atom on the surface to make it completely hydroxylated, so that the constructed rock surface conformed to the actual situation where the shale pore was hydrophilic [37,38,39]. The surface hydroxyl density was 9.6 nm^−2^, which was consistent with the results of the crystal chemistry calculations (5.9–18.8 nm^−2^) [40]. In addition, the effect of surface hydroxyl density in this study was investigated using some simple characterizations in the Appendix A. The length of the rock surface in the Z-direction was 103.5 Å, the thickness in the Y-direction was 28.0 Å, and the widest point of the rock surface in the X-direction was 23.4 Å. The position of the silica surface was fixed. The narrowest point of the wedge-shaped nanopores was 30.0 Å and the widest point was 80.0 Å. Octane was used to represent shale oil during the displacement simulation in the medium, in which 517 molecules were randomly placed in single pore and 517 molecules were randomly placed in both of the double pores, with a total of 1034 octane molecules. To ensure that the results of this study were accurate, we also chose dodecane to represent the oil phase. Details are given in the Appendix A. Figure 8a,b show the wedge-shaped single pore Ⅰ ranging in size from 8 nm to 3 nm, and single pore Ⅱ from 3 nm to 8 nm, respectively, and Figure 8c shows wedge-shaped double pores Ⅰ and Ⅱ. To compare with wedge-shaped pores, 3 nm and 8 nm slit-shaped pores were constructed, and the details can be found in the Appendix A. Prior to the CO_2_ flooding simulation, a 3 ns equilibrium molecular dynamics simulation was performed to obtain a reasonable oil density at 333 K and 20 MPa based on the reservoir conditions. The displacement simulation was based on non-equilibrium molecular dynamics (NEMD). The NEMD simulations that were run for the wedge-shaped pores and 3 nm slit-shaped pore systems were 5 ns, and the 8 nm slit-shaped pore system was 2 ns. During the CO_2_ flooding simulation, a He plate was placed on the left and right sides, and different pressures were applied to the plates to generate a pressure difference that allowed CO_2_ to be injected. The pressure applied to the left side was 25 MPa, and the pressure applied to the right side was 20 MPa.

### 3.2. Simulation Details

For all of the models, periodic boundary conditions were used. With time step of 1 fs, all MD simulations were carried out in NVT system synthesis [41] by using the LAMMPS (Large-scale Atomic/Molecular Massively Parallel Simulator) program. The thermostat was a Nosé–Hoover [42] set to 333 K to regulate the temperature. The He plates and rock surfaces were regarded as rigid bodies. Octane molecules were modeled by using the OPLS-AA force field [43], CO_2_ was modeled by using the EMP2 force field [44,45], and the quartz rock surface was modeled by using the ClayFF force field [46]. The densities of oil and CO_2_ in the bulk phase state were computed and compared with the information in the standard database of the National Institute of Standards and Technology (NIST) in order to confirm the precision of each force field in our implemented system. The comparison results are shown in the Appendix A, where barely any density differences can be seen, indicating that the force fields we used in this study were appropriate. Utilizing the Ewald summation method [47,48], the long-range electrostatic interaction was calculated between two atoms, *i* and *j*, as the sum of the Lenard–Jones (LJ) and electrostatic potential energies [49]:(3)$$E_{ij}=\frac{q_{i}q_{j}}{r_{ij}}+4\varepsilon_{ij}\left[{\left(\frac{\sigma_{ij}}{r_{ij}}\right)}^{12}-{\left(\frac{\sigma_{ij}}{r_{ij}}\right)}^{6}\right]$$

With a cut-off radius of 10.0 for the non-bonding interaction and a non-bonding energy $E_{ij}$ as the result, where $r_{ij}$ stands for the distance between two atoms *i* and *j*, $q_{i}$ and $q_{j}$ stand for the charges of those atoms, respectively, and $\varepsilon_{ij}$ and $\sigma_{ij}$ stand for the LJ trap depth and LJ radius between atoms *i* and *j*, respectively. The Lorentz–Berthelot rule governed how various atom types interact with one another [50]:(4)$$\sigma_{ij}=\frac{1}{2}\left(\sigma_{ii}+\sigma_{jj}\right)$$
(5)$$\varepsilon_{ij}={\left(\varepsilon_{ii}\varepsilon_{jj}\right)}^{\frac{1}{2}}t$$

## 4. Conclusions

In this work, CO_2_ flooding in four types of wedge-shaped pores, together with two types of slit-shaped pores were investigated by using molecular dynamics simulations. The main conclusions are as follows:

(1) In wedge-shaped pores, CO_2_ transport is much faster in single pore Ⅱ and double pore Ⅱ than that in single pore Ⅰ and double pore Ⅰ, and this flow velocity difference is more pronounced between double pores. During the oil displacement process, some oil molecules in wedge-shaped single pore Ⅰ and double pore Ⅰ diffuse into the left side of pores, and then oil molecules re-enter the wedge-shaped pores together with CO_2_, while this phenomenon does not appear in wedge-shaped single pore Ⅱ and double pore Ⅱ. 

(2) In wedge-shaped pores, the “oil/CO_2_ inter-pore migration” mechanism is proposed according to the migration of oil/CO_2_ molecules from double pore Ⅰ to Ⅱ and the center of the mass analyses. The interaction and RDF analyses confirm that the oil has more contact with CO_2_, a better miscible phase, greater interaction, and faster displacement in the wide inlet in single pore Ⅰ and double pore Ⅰ, as well as in the wide outlet in single pore Ⅱ and double pore Ⅱ. 

(3) The efficiency of the whole oil displacement process follows the sequence of 8 nm > double pore Ⅱ > single pore Ⅱ > 3 nm > double pore Ⅰ > single pore Ⅰ. The inlet and outlet of the pores have an influence on oil displacement, and the restriction influence from the outlet is much greater than that from the inlet.

## Figures and Tables

**Figure 1 molecules-28-00188-f001:**
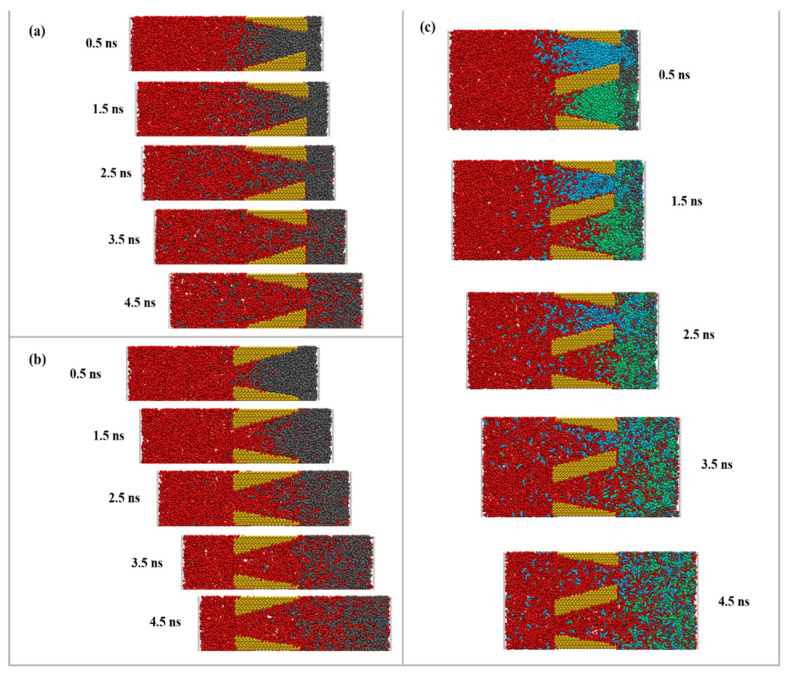
Snapshots to indicate the evolutions of CO_2_ flooding in (**a**) single pore Ⅰ; (**b**) single pore Ⅱ; (**c**) double pore.

**Figure 2 molecules-28-00188-f002:**
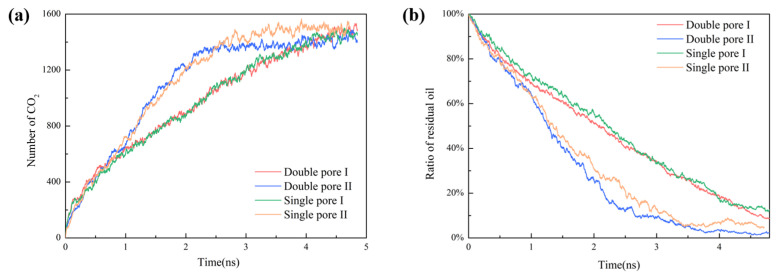
(**a**) The number of CO_2_ molecules entering in each wedge-shaped pore and (**b**) the ratio of residual oil.

**Figure 3 molecules-28-00188-f003:**
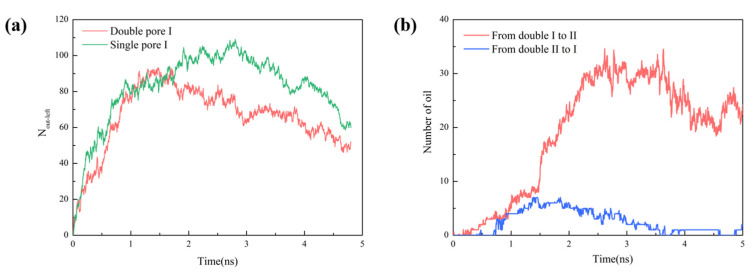
(**a**) The number of oil molecules on the left side of double pore Ⅰ and single pore Ⅰ; (**b**) number of oil molecules migrating from double pore Ⅰ to double pore Ⅱ and from double pore Ⅱ to double pore Ⅰ.

**Figure 4 molecules-28-00188-f004:**
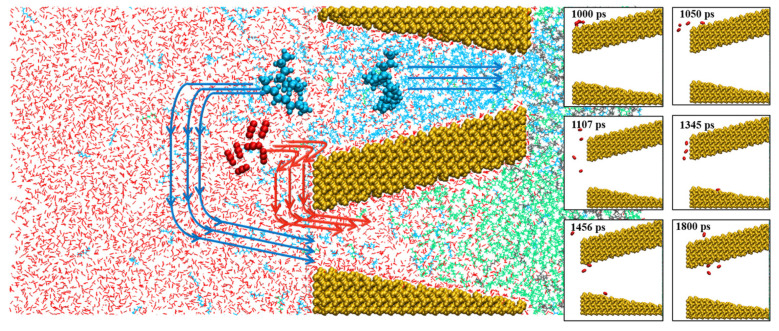
Left side: streamlines of oil molecules/CO_2_ fluid migration from double pore Ⅰ to double pore Ⅱ. Right side: snapshot of CO_2_ migration from double pore Ⅰ to double pore Ⅱ at different times.

**Figure 5 molecules-28-00188-f005:**
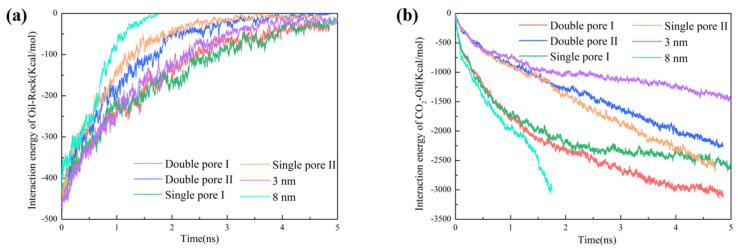
Interaction energy between (**a**) oil and rock surface, and (**b**) CO_2_ and oil during oil displacement at 333 K.

**Figure 6 molecules-28-00188-f006:**
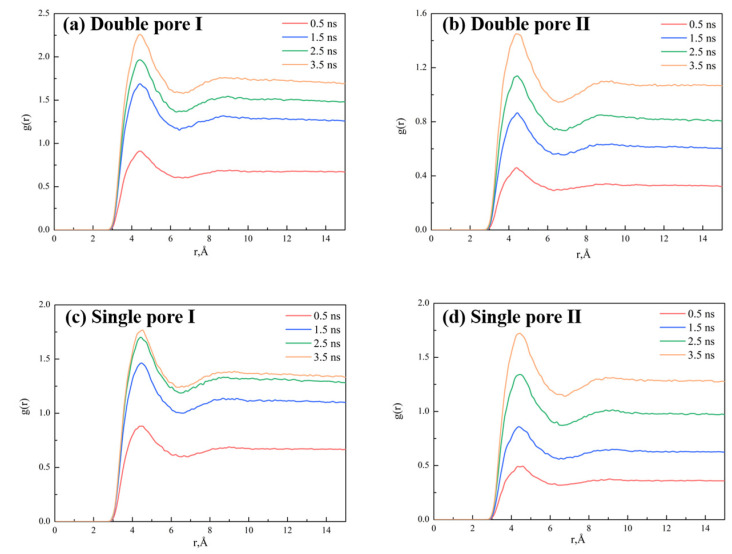
Radial distribution function g(r) between C (CO_2_)-C (oil) atoms in (**a**) double pore Ⅰ, (**b**) double pore Ⅱ, (**c**) single pore Ⅰ, and (**d**) single pore Ⅱ.

**Figure 7 molecules-28-00188-f007:**
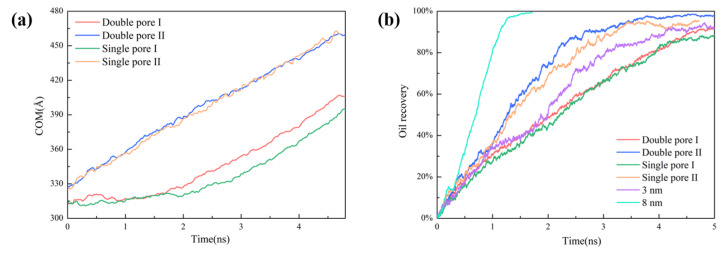
(**a**) COM of oil with time, and (**b**) oil recovery in different pore structures.

**Figure 8 molecules-28-00188-f008:**
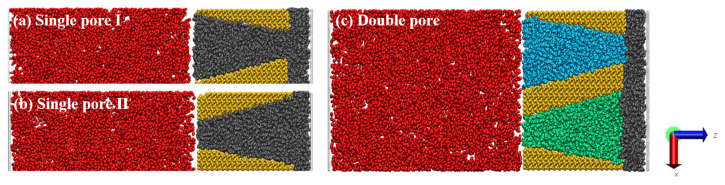
(**a**) Wedge-shaped single pore Ⅰ with 8 to 3 nm sized pores; (**b**) wedge-shaped single pore Ⅱ with 3 to 8 nm pore; (**c**) wedge-shaped double pore, pore size 3 to 8 nm for double pore Ⅰ, 8 to 3 nm for double pore Ⅱ (red: CO_2_ molecule; gray, cyan, green: oil phase; orange: shale rock surface).

## Data Availability

Not applicable.

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
