# Peer review of "Molecular Dynamics Insight into the CO2 Flooding Mechanism in Wedge-Shaped Pores"

_molecules, 2022, doi:10.3390/molecules28010188_

Round 1
Reviewer 1 Report
This article investigated the oil displacement by CO2 in wedge-shaped pores via MD simulation. The results confirmed the wedge-shaped pore Ⅱ left nonnegligible effects on the oil displacement efficiency compared with pore Ⅰ. The “oil/CO2 inter-pore migration” and “siphoning” phenomena could also be observed in wedge-shaped double pores with larger CO2-oil contact surface, better phase miscibility, and faster displacement. My suggestion is to accept after minor revision, and the author should answer the following four questions.
1. In section 2.1, the article pointed the silica surface were completely hydroxylated. Whereas, the silica surface was particle hydroxylated in actual situation, which was studied in this article: https://pubs.acs.org/doi/10.1021/cm500365c. To make the results more credible, it is suggested that the authors consider the effect of surface hydroxyl density on this study by attaching some simple characterizations to the supporting material.
2. In Figure 3, there was little difference in the CO2 number entered by two single pore Ⅰ&Ⅱ with different structures. However, the author did not explain the reason for this result later.
3. The right side of Figure 5 showed that the number of CO2 molecules passing along the wall is finite. We thought that the main effect of porous CO2 in Figure 3a is CO2 bulk phase increased, which could not form a strict contrast with the single pore.
4. In the line 192-193, the authors thoughts the oil molecules in double pore Ⅰ were hindered by the pore surface. It could be seen from the left of Figure 5 that the competitive adsorption of CO2 and oil molecules greatly reduces the hinder effects of the wall on oil molecules. The authors should consider the pressure difference between the inlet and outlet locations due to the difference in fluid velocity, which was likely to be the main factor leading to the migration of oil molecules.
Reviewer 2 Report
In this manuscript, the CO2 flooding mechanism was studied by molecular dynamics simulation. After reviewing the manuscript, some suggestions arise as follows:
1. The manuscript needs careful modifications, such as English grammar and sentence structures.
2. In recent years, many scientific papers have reported the alkane displacement from hydrophilic silica surfaces by experiment and simulation methods. What are the advantages of the CO2 flooding compared with the hot-warm flooding and surfactant flooding? In the tertiary recovery, most solid surfaces are hydrophobic in the reservoir. It is more important to study the oil displacement at the hydrophobic surfaces.
3. The composition of crude oil is complex and varies greatly, thus, the Octane is not representative. Maybe the authors can try to study some different oil samples to see if they come to similar conclusions.4. In “2. Models and methodology” section, the position of silica surface is not fixed. In “Figure 2”, the positions of silica surface are different as the function of time. In the previous works, the coordinates of solid surfaces are fixed during the simulation process. The movements of silica surface might lead to the “oil /CO2 inter- pore migration".
5. In “2.2 Simulation details” section, how long had the MD simulations run for the different systems?
6. In “Figure 6”, it should be “interaction energy” in both figure caption and the y-axis.
Round 2
Reviewer 2 Report
As the reviewer suggested, we also changed the octane in the double pore calculation to dodecane[7-8] and discovered that the overall phenomena were compatible with the same conclusions as when octane represented the oil phase. (See Supporting Information "1. Different types of alkanes represent the oil phase")
Octane and dodecane are both saturates. The aromatics, resins, and asphaltenes in crude oil are more important for the enhanced oil recovery and they are more difficulty to displace from the solid surface.
Author Response
Revisions and replies to the comments of the reviewers
(molecules-2090814)
Reviewer #2:
Comments:
Point 1: Octane and dodecane are both saturates. The aromatics, resins, and asphaltenes in crude oil are more important for the enhanced oil recovery and they are more difficulty to displace from the solid surface.
Response 1: We show our great appreciation to the reviewer for the suggestions. Numerous researches [1–7] have employed saturated alkanes in their publications, demonstrating that it is viable to use saturated alkanes to represent the oil phase. Of course, crude oil contains aromatics, resins, and asphaltenes, they interact more strongly with the solid surface and are more difficult to displace from the solid surface, which is more important for enhanced oil recovery. As a result, we will take this issue into account in our future work.
Furthermore, the effect of wedge-shaped pores structure is primarily considered in this study, and the oil phase used in all structures is the same, so the presence of aromatics, resins, and asphaltenes in the oil phase does not affect the conclusions of this work.
- Fang T, Zhang Y, Ma R, et al. Oil extraction mechanism in CO2 flooding from rough surface: Molecular dynamics simulation. Applied Surface Science, 2019, 494, 80-86; DOI: 1016/j.apsusc.2019.07.190.
- Fang C, Yang Y, Sun S, et al. Low salinity effect on the recovery of oil trapped by nanopores: A molecular dynamics study. Fuel, 2020, 261, 116443; DOI: 1016/j.fuel.2019.116443.
- Xu S, Wang J, Wu J, et al. Oil contact angles in a water-decane-silicon dioxide system: effects of surface charge. Nanoscale research letters, 2018, 13, 1-9; DOI: 1186/s11671-018-2521-6.
- Lu P, Mo T, Wei Y, et al. Molecular insight into oil displacement by CO2 flooding on rough silica surface. The Journal of Supercritical Fluids, 2022, 181, 105507; DOI: 1016/j.supflu.2021.105507.
- Yan Y, Dong Z, Zhang Y, et al. CO 2 activating hydrocarbon transport across nanopore throat: insights from molecular dynamics simulation. Physical Chemistry Chemical Physics, 2017, 19, 30439-30444; DOI: 10.1039/c7cp05759h.
- Fang T, Zhang Y, Ding B, et al. Static and dynamic behavior of CO2 enhanced oil recovery in nanoslits: Effects of mineral type and oil components. International Journal of Heat and Mass Transfer, 2020, 153, 119583; DOI: 10.1016/j.ijheatmasstransfer.2020.119583.
- Liu B, Liu W, Pan Z, et al. Supercritical CO2 breaking through a water bridge and enhancing shale oil recovery: A molecular dynamics simulation study[J]. Energy & Fuels, 2022, 36, 7558-7568; DOI: 1021/acs.energyfuels.2c01547.
Thanks again for your and the reviewers’ careful work! Should you need any further information, please do not hesitate to contact us. Many thanks for your time and consideration.
Best regards!
